# Experience of irreproducibility as a risk factor for poor mental health in biomedical science doctoral students: A survey and interview-based study

**Nasser Lubega[1], Abigail Anderson[2], Nicole C. Nelson** [1] *

**1** University of Wisconsin–Madison School of Medicine and Public Health, Madison WI, United States of America, **2** Midwestern University Chicago College of Osteopathic Medicine, Chicago IL, United States of America

* nicole.nelson@wisc.edu

**Data Availability Statement:** The survey data, interview data, and code used to generate the figures are available at: https://github.com/nicole-c-nelson/repro_MH.

## Abstract

High rates of irreproducibility and of poor mental health in graduate students have been reported in the biomedical sciences in the past ten years, but to date, little research has investigated whether these two trends interact. In this study, we ask whether the experience of failing to replicate an expected finding impacts graduate students' mental health. Using an online survey paired with semi-structured qualitative interviews, we examined how often biomedical science doctoral students at a large American public university experienced events that could be interpreted as failures to replicate and how they responded to these experiences. We found that almost all participants had experience with irreproducibility: 84% had failed to replicate their own results, 70% had failed to replicate a colleague's finding, and 58% had failed to replicate a result from the published literature. Participants reported feelings of self-doubt, frustration, and depression while experiencing irreproducibility, and in 24% of cases, these emotional responses were strong enough to interfere with participants' eating, sleeping, or ability to work. A majority (82%) of participants initially believed that the anomalous results could be attributed to their own error. However, after further experimentation, most participants concluded that the original result was wrong (38%), that there was a key difference between the original experiment and their own (17%), or that there was a problem with the protocol (17%). These results suggest that biomedical science graduate students may be biased towards initially interpreting failures to replicate as indicative of a lack of skill, which may trigger or perpetuate feelings of anxiety, depression, or impostorism.

## Introduction

Two conversations about systemic problems in science have emerged over the past decade: one focused on the reproducibility and rigor of published findings, and another on the mental health of graduate students. While the problem of irreproducibility is not new, the scale of the problem was until recently underappreciated. Reports from pharmaceutical companies of

**Funding:** NL received funding from the Shapiro summer research program at the University of Wisconsin Madison (https://summerresearch.med. wisc.edu/). AA received funding from the undergraduate research opportunity program at the Holtz Center for Science and Technology Studies at the University of Wisconsin Madison. NCN received no specific funding for this study.

**Competing interests:** The authors have declared that no competing interests exist.

replication rates as low as 11–39% in preclinical research [1,2] have catalyzed discussion about reproducibility and rigor in the past decade [3]. This growing literature has focused on identifying the specific types of problems that may give rise to irreproducibility (such as publication bias or lack of masking) [4,5], quantifying how often these problems arise [6–8], and proposing and evaluating potential solutions [9–11].

At the same time, reports of high rates of anxiety and depression in graduate students have sparked a growing interest in student mental health. Reports from the University of California Berkeley [12] and the University of Arizona [13] that nearly half of graduate students in the biomedical sciences were depressed or reported higher than average stress levels prompted calls for reform [14–16]. Subsequent, more extensive surveys have reported levels of depression and/or anxiety in PhD students that are 1.5 to 6 times that of comparable populations [16–18], with estimated prevalence rates generated through meta-analyses ranging from 17 to 35% [19,20]. Recent analyses have shown that wellness and mental health is a small but growing area of investigation in the literature on biomedical graduate education [21], and research on graduate student mental health more generally has increased especially sharply in the past two years (probably owing to the Covid-19 pandemic) [22]. A recent systematic review of this field has shown that research has focused on primarily on identifying correlates of poor mental health (such as concerns about career prospects or work-life balance), coping strategies/interventions and protective factors (such as strong mentoring relationships and social networks), and disentangling whether graduate-level study is a cause of poor mental health or is simply correlated [23].

To date, virtually no research has asked whether or how these two trends interact. Research on reproducibility and rigor has tended to focus on the science rather than the scientists [21]. Conversely, research on STEMM graduate education has tended to focus on the scientists and the institutional structures in which they work, but not on the content of scientific work itself. Some commentaries have posed the question of whether conducting formal replications or adopting more rigorous practices might harm the career prospects of early career researchers [24–26], but little is known about this question or any other ways that irreproducibility might impact students. Anecdotal evidence, however, suggests a link between the experience of failing to replicate a result and mental health: In a *Nature* column, one biologist poignantly described how concerns about the replicability of his research triggered a depressive episode of his own during his PhD studies [27].

Our ability to study potential relationships between reproducibility and student mental health is also limited by a relative lack of data on how often scientists experience failures to replicate and how they respond. While several large-scale projects have attempted to quantify the proportion of published findings that are replicable in different fields [28,29], only a few surveys have attempted to quantify the proportion of scientists who have failed to replicate a result [30–32]. A 2019 National Academies report noted that this existing research on irreproducibility experiences relies primarily on convenience samples, limiting the strength of the conclusions that can be drawn from it [33]. Existing surveys have also relied on respondents to identify an experience as a failure to replicate. Because of the difficulty of distinguishing between a true null finding, a malfunctioning experimental system, and a lack of experimenter skill [34], not all respondents may identify their own experiences with terms such as "irreproducibility" or "failure to replicate."

In this study, we used a combination of survey and semi-structured interview techniques to assess how often graduate students in the biomedical sciences at a large American public university experience situations that could be framed as irreproducibility, how they respond to these experiences, and whether those responses might impact their mental health. We used scenario-based survey questions to assess students' experiences with a range of situations,

followed by qualitative interviews that allowed students to elaborate in detail on one or more of their experiences with irreproducibility.

## Materials and methods

### Participants and recruitment

We recruited participants from training grant programs in the biomedical sciences at the School of Medicine and Public Health at the University of Wisconsin–Madison. National Institute of General Medical Sciences (NIGMS) training grants award PhD students with guaranteed funding for up to five years, and only US citizens and permanent residents are eligible for these grants. A total of 126 students in four training grant programs were invited by email to participate in our study in Spring 2020. We first invited students to complete a survey that collected demographic information, assessments of irreproducibility experiences, and current mental health. Participants were informed through the online consent process that the aim of the study was to examine possible relationships between irreproducibility and student mental health. The survey was administered using Qualtrics survey software. For those who agreed to participate, the median time to complete the survey was five minutes and 20 seconds (min 2:00, max 17:43:16). The complete survey is available as S1 File. Participants whose responses suggested that they had experienced irreproducibility were then invited by email to participate in a semi-structured interview.

### Ethics approval

The study was reviewed by UW Madison's Education and Social/Behavioral Science IRB and determined to meet the criteria for exempt human subjects research under 45 CFR 46 Category 2: Research involving the use of educational tests, surveys, and interviews (Submission ID#: 2020–0522).

### Demographic information

We asked participants to identify their 1) gender (Man, Woman, Transgender, Nonbinary, Other), 2) ethnicity (Hispanic/Latinx or not), and 3) race (American Indian/Alaskan Native, Asian, Black/African American, Native Hawaiian/other Pacific Islander, White, Other). Multiple responses were enabled so that participants could identify with more than one gender or racial category. In accordance with NIH definitions, we classified participants as belonging to a racial/ethnic group that is under-represented in American biomedicine if they identified as Hispanic/Latinx, American Indian/Alaskan Native, Black/African American, or Native Hawaiian/other Pacific Islander [35]. Participants were also asked if they came from a disadvantaged background, defined by the NIH as meeting two or more of the following criteria: A) Were or currently are homeless, B) Were or currently are in the foster care system, C) Were eligible for the Federal Free and Reduced Lunch Program for two or more years, D) Do not have a parent/legal guardian who completed a bachelor's degree, E) Were or currently are eligible for Federal Pell grants, F) Received support from the Special Supplemental Nutrition Program for Women, Infants, and Children (WIC) as a parent or child, G) Grew up in a rural area or a low income/health professional shortage area [35].

### Current mental health

We assessed participants' current mental health using the PHQ-8, GAD-7, and the General Life Satisfaction Fixed Form B scales. The PHQ-8 is a widely used depression screening tool with well-validated cutoff criteria for scores suggestive of clinical depression (50). It assesses

respondents' experiences of the DSM-IV criteria for depression over the last two weeks using a four-point Likert scale (ranging from "Not at all" to "Nearly every day"). The GAD-7 is a widely used anxiety screening tool that similarly uses a four-point Likert scale to screen for probable Generalized Anxiety Disorder (51). The General Life Satisfaction Fixed Form B is a newer scale developed as part of the NIH Toolbox measures to assess psychological well-being (52). This scale was developed and validated based on a nationally representative sample of American adults and focused on respondents' cognitive evaluation of their life experiences (rather than their current affective state).

## Experiences of irreproducibility

We assessed participants' potential experiences of irreproducibility through descriptions of experimental scenarios that could be interpreted as failures to replicate. We asked participants: Have you experienced a situation where you were doing an experiment that A) gave you that results were not consistent with what you expected, B) had a "right answer," and you did not get that answer, C) you yourself had done before, but got results that differed from your previous attempts, D) someone else in your lab had done before, but you got results that differed from theirs, E) was similar to an experiment you'd seen in the published literature, but the results you got differed from what was reported in the publication, F) had a control group or comparison group that was supposed to produce an expected result, but you got a different result. Participants were instructed to consider all their scientific experience to date when answering the questions, including lab work in courses, internships, or volunteer positions. For each question, participants could indicate that they had experienced such a scenario A) one time, B) several times, C) never, or D) not sure. Participants were invited to participate in the interview phase of the study if they indicated that they had experienced two or more of these scenarios at least one time.

## Comparison with the Healthy Minds Study data

Sampling bias and selection bias are perennial problems in survey research. In this study, we were especially concerned about whether our participants had significantly higher or lower rates of mental illness compared to the broader population of STEMM PhD students at American universities. To assess the representativeness of our participants, we compared our data with data from the Healthy Minds Study (HMS). The HMS is an annual web-based survey examining mental health status and service utilization in undergraduate and graduate students at post-secondary institutions across the United States. Because of the mental health impacts of the Covid-19 pandemic, we used HMS survey data collected during the same academic year that our study was conducted (2019–2020) so that our current mental health measurements would be comparable. In addition to the large sample size, randomly sampled, and nationally representative nature of the HMS survey, the HMS study corrects for non-response bias by comparing the demographic characteristics of participants to the demographic composition of their home schools and constructing a weight for each participant. The less likely a particular type of student was to complete the survey, the more weight they received in the analysis. We extracted these response propensity weights, demographic information, PHQ-8 scores, and GAD-7 scores for students enrolled in PhD programs in the natural sciences, engineering, medicine, nursing, pharmacy, or public health (excluded PhD fields were humanities, social sciences, architecture/urban planning, art/design, business, education, law, music/theatre/ dance, public policy, and social work). HMS participants were classified as belonging to a racial/ethnic group that is underrepresented in American biomedicine in accordance with the NIH definition described above. We performed one-sample proportion tests to look for

statistically significant differences between our participants and the response-weighted HMS reference population in the proportion of women, participants from under-represented groups, participants with probable anxiety, and participants with probable depression.

## Interviews

Participants were individually interviewed via Cisco WebEx video conferencing by one of our five study team members (two white non-Hispanic women, one Black man, one white non-Hispanic man, and one white Hispanic man). Thirty-six of 37 interviews were performed by study team members who were undergraduate or medical students so that social status discrepancies were less likely to influence participants' disclosure of information. Participants had no relationship to study team members prior to the study. Study team members used an interview guide to ensure consistency in the data collected by each team member (S2 File) and trained in administering the interview guide by conducting mock interviews and analyzing recordings of those mock interviews before performing interviews with participants. The interview questions asked participants to recount details of their irreproducibility experiences, steps they took after identifying the problem, thoughts about the source of the irreproducibility, and the impact of these experiences. To avoid question order bias, the guide was structured so that study team members first asked open-ended questions about the impact the participant's reproducibility experience had on them and then prompted participants to speak specifically about impacts on their mental health, enthusiasm for science, and career progression. When participants had more than one experience with irreproducibility, interviewers asked participants to focus on the experience which had the strongest impact on them and collected data on additional experiences if time permitted. The average length of the interviews was approximately 30 minutes. Interviews were recorded, and audio was extracted, transcribed to a clean verbatim standard, double-checked for accuracy, and de-identified (i.e., specific names and details were replaced with nonspecific identifiers such as "supervisor," "lab," or "gene X"). Transcripts were not given to participants for review, and no repeat interviews were performed. All participants were invited to attend a presentation and provide feedback on the findings.

## Analysis of survey data

Our primary aim in collecting the survey data was to assess how frequently students may have experienced different kinds of irreproducibility and to assess whether our survey participants were representative of the broader population of STEMM PhD students at American universities. While it would be possible to assess correlations between past experiences of irreproducibility and current mental health measures, we have elected not to do so because our scenario-based questions about irreproducibility are not a validated measure of exposure to irreproducibility, and because the length of time elapsed between an irreproducibility experience and the current mental health data captured in the survey varied greatly between participants. Consequently, we use the demographic and current mental health data only to assess selection bias, and report only the raw survey data on experiences of irreproducibility.

## Analysis of interview data

To assess possible selection bias in the participants who responded to the invitation to participate the interview phase of our study, we conducted chi squared tests to see if there were significant differences in demographics and the proportion of people with probable depression or anxiety in our survey and interview samples, and two-sided t-tests to assess differences in PHQ-8, GAD-7, and GLS scores. For the qualitative data analysis, three study team members developed a coding scheme to analyze the transcripts using a content analysis approach (53).

All three of these study team members had performed at least one interview during data collection. Participants were identified only by a numeric study identifier during the qualitative data analysis phase to conceal participant identity from study team members. These four digit numeric identifiers are also used in this manuscript to allow the reader to identify quotes from unique participants in the data set. Participants' demographic data—which was collected through the survey instrument and not in the interviews—was linked to the participants' interviews only after qualitative data analysis was complete. The coding scheme included three types of items: 1) categorizations of elements of the participant's experience (e.g., type of irreproducibility or emotional response), 2) presence or absence of elements in a participant's story (e.g., mentioning external support networks or receiving treatment for a mental health issue), and 3) three-point scale assessments of the participant's experience (e.g., positive/neutral/negative relationship with advisor). The complete coding scheme is available in S3 File. Each of the same three study team members coded all transcripts in full using this coding scheme. Coding inconsistencies were resolved through discussion: for items where scores diverged, study team members identified specific lines of the transcript that influenced their coding and came to an agreement about how best to interpret and weigh the evidence from these passages. The survey data, interview data, and code used to generate the figures are available at: https://github.com/nicole-c-nelson/repro_MH. A completed COREQ checklist is available as S4 File.

## Results

Of the 126 students invited, 80 completed the survey. Based on their survey responses, 79 participants were invited for a follow-up interview and 37 participants completed that interview. Demographic characteristics and measures of current mental health for our survey and interview participants are presented in Table 1. To contextualize our participants in the broader

**Table 1. Demographic characteristics and mental health measures for survey, interview, and Healthy Minds Study samples.**

|  | Survey (n = 80) | Interview (n = 37) | HMS raw data (n = 3902) | HMS weighted data |
|---|---|---|---|---|
| **Gender identity** |  |  |  |  |
| Man | 56% (n = 45) | 76% (n = 28) | 44% (n = 1704) | 55% |
| Woman | 44% (n = 35) | 24% (n = 9) | 54% (n = 2113) | 43% |
| Trans/non-binary/genderqueer | 0% (n = 0) | 0% (n = 0) | 2% (n = 78) | 2% |
| **Under-represented racial/ethnic group** |  |  |  |  |
| Yes | 23% (n = 18) | 16% (n = 6) | 13% (n = 516) | 13% |
| No | 78% (n = 62) | 84% (n = 31) | 86% (n = 3366) | 86% |
| **Socioeconomic background** |  |  |  |  |
| Disadvantaged | 21% (n = 17) | 24% (n = 9) |  |  |
| Not disadvantaged | 79% (n = 63) | 76% (n = 28) |  |  |
| **GAD7 (range 0–21)** |  |  |  |  |
| Mean | 5.58 (SD = 4.40) | 5.70 (SD = 4.47) | 6.35 (SD = 5.19) | 6.19 (SD = 5.17) |
| Probable anxiety ($> = 10$) | 19% (n = 15) | 19% (n = 7) | 23% (n = 892) | 21% |
| **PHQ8 (range 0–24)** |  |  |  |  |
| Mean | 5.49 (SD = 4.19) | 5.54 (SD = 4.42) | 7.18 (SD = 5.28) | 7.04 (SD = 5.31) |
| Probable depression ($> = 10$) | 13% (n = 10) | 8% (n = 3) | 26% (n = 1025) | 26% |
| Probable anxiety and depression | 10% (n = 8) | 8% (n = 3) | 16% (n = 613) | 15% |
| **General Life Satisfaction (range 5–25)** |  |  |  |  |
| Mean | 19.3 (SD = 3.94) | 19.8 (SD = 3.79) |  |  |
| Low GLS ($> = 14$) | 15% (n = 12) | 16% (n = 6) |  |  |

population of science, technology, engineering, mathematics, and medicine (STEMM) PhD students in the United States, we compared our survey data to demographic and current mental health data collected in the Healthy Minds Study (HMS) during the same academic year (2019–2020). Our survey had a significantly higher proportion of people from under-represented groups compared to the weighted HMS sample ($c^2(1) = 5.31$, $p = .021$), but the difference in the proportion of women versus men was not significant. The proportion of people with probable moderate to severe depression in our survey was significantly lower than in the HMS survey population ($c^2(1) = 6.519$, $p = .011$). The difference in the proportion of people with probable Generalized Anxiety Disorder (GAD) between our survey and HMS data was not significant. None of the differences between the survey and interview participants in the proportion of women, people from under-represented groups, or people from disadvantaged backgrounds were significant. Survey/interview differences in mean GAD-7, PHQ-8, and GLS scores, and in the proportion of people with probable GAD or depressive disorders were also not significant.

Nearly all participants (99%) indicated in our survey that they had experienced at least two of the irreproducibility scenarios we described (Fig 1). Eighty-four percent of participants reported at least one experience with failing to replicate their own results, 70% had failed at least once replicate work from a colleague in their laboratory, and 58% had failed at least once to replicate a result from the published literature. Participants expressed the greatest uncertainty about how to answer the questions on replicating results from the published literature, from lab members, and in situations where there was an expected "right" answer. Based on the experiences participants described in the follow-up interviews, we hypothesize that ambiguity around who counts as a lab member or whether an experience counts as a replication of published work if a former lab member produced that work may account for some of the uncertainty expressed in participants' responses to those survey questions.

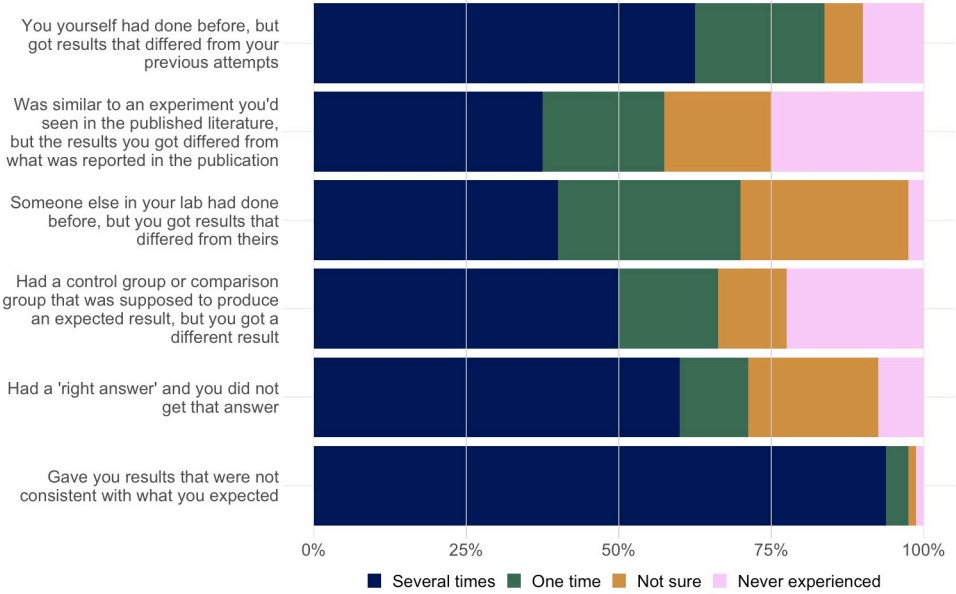

**Fig 1. Responses to survey questions about potential irreproducibility scenarios.** Responses were recorded from all 80 survey participants.

We classified the irreproducibility experiences that participants described in follow-up interviews as falling into three categories: failures to replicate a result from the literature (29%), failures to replicate their own work (24%), and failures to replicate work performed by someone related to their lab (48%). The first category included both exact/direct and conceptual replications of published studies; that is, both attempts to reproduce the results using the same materials and methods as the original and attempts to test the fundamental hypothesis of the original study (e.g., to test whether cancer cells are sensitive to a particular compound using a different cell line than the original authors) [36]. The last category included attempts to replicate published or unpublished results from a former lab member or close collaborators who were not directly in the participants' laboratory. What distinguished these experiences from attempts to replicate a result from the literature was that the participants had some form of social relationship with the person who produced the original results, which provided a higher degree of access to tacit knowledge about the methodology as well as knowledge of the experimenter's reputed level of skill.

While most interview participants chose to speak about an experience that had taken place during their graduate training (71%), a sizeable minority spoke about an irreproducibility experience from their undergraduate or pre-doctoral training (29%). Participants with multiple irreproducibility experiences described their early experiences as more impactful because they tended to place more trust in the literature as beginning graduate students and had fewer data points about their own competencies. As students progressed and had success in other projects, they became less likely to blame themselves when experiencing irreproducibility and more likely to question the existing literature. Participants also reported more success in convincing their PIs of their interpretation of a problem when they were more advanced students. As one student put it, "I think part of why I was less troubled by this [irreproducibility experience] was that it came along last year or so of my graduate career, and at that point, my committee knew that I could do stuff. They weren't concerned that I didn't have hands or something, or if I was just procrastinating. It wasn't a reflection of me that I couldn't get this to work because I could get other stuff to work. Had I been a first-year student, potentially, I may have had more self-doubt that I couldn't get stuff to work" (1006).

Fig 2 details trajectories for a subset of irreproducibility experiences (n = 22) where we were able to clearly distinguish between the participant's initial thoughts about why they were unable to obtain the expected results and their thoughts at the time of interview. Eighty-two percent of these participants initially assumed that their unexpected results were due to their lack of knowledge/skill or an objective error on their part (e.g., incomplete removal of ethanol following DNA precipitation or ordering the wrong primer sequence). These attributions changed markedly as participants continued working with their experimental system. At the time that we interviewed them, none of the participants in this subset believed that the failure to replicate was due to their own error, and half of participants now believed that the original result that they had been attempting to replicate was incorrect. Others now attributed their failure to replicate to a problem with the protocol or a bad reagent (e.g., nonspecific antibody binding), a key difference between the original and replication experiment (e.g., species differences), or to the variation inherent in complex biological systems.

In addition to further experimentation, participants reported that mentorship from PIs played an important role in how their attributions changed over time. In some cases, PIs helped reframe participants' initial perceptions that they were at fault and suggested alternative explanations. One recalled, "I talked to my advisor, and it ended up being a positive experience because she trusted the result. She said, 'I don't think you did anything wrong. I think this is real.' It was nice because I think it really validated my skill and identity as a researcher" (1029). Other participants found themselves in conflict with their PIs as they began to disbelieve the

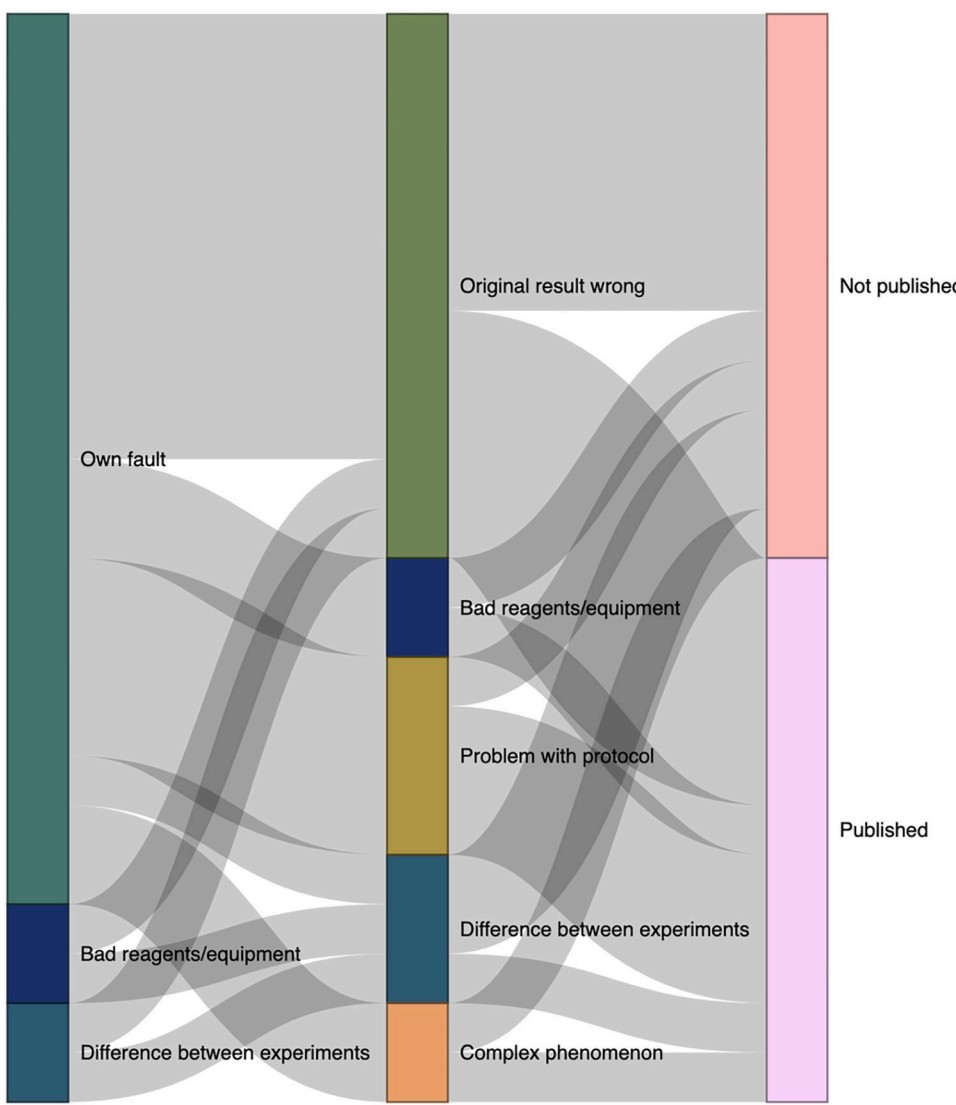

**Fig 2. Sankey diagram of the trajectory of irreproducibility experiences (n = 22).** Column one represents participants' initial assumptions about the reason their experiments produced anomalous results, and column two represents participants' assumptions at the time of the interview. Column three indicates whether data from these experiments was eventually published. The thickness of the lines connecting the columns is proportional to the number of experiences.

original result after further experimentation, but their PI persisted in framing the situation as one of student error. As one student described, "[My PI's] first instinct is like, 'Oh, well, maybe you messed up this prep.' It's like, no, I didn't. [My PI] continually suggest[ed] it's the prep's fault or I left ethanol after I washed; it has not been that case once this entire time" (1056). For replications of results produced within the extended lab group, the PI's perception of the person who originally produced the results played an important role in how the PI responded to the potential failure to replicate. One student recalled that her PI was unsurprised that her results differed from those obtained by the previous student because that student was "notoriously a mess in the lab" (2009).

Half of the participants in this subset reported that results from their experiments had been published or were being written up for publication at the time of the interview. There was no

relationship between the participant's current attribution and the likelihood of publication—in almost all attribution categories, participants were equally likely to report publishing or not publishing their results. It is worth noting, however, that this does not mean that the participants published conflicting findings or information about sources of irreproducibility in all cases. Some participants successfully found alternative conditions under which they could produce a similar result or used an alternative technique to arrive at a similar conclusion, and then published results related to their research goals without reporting on the problems they experienced with the originally published method.

In the complete set of irreproducibility experience interviews (n = 37 participants, n = 42 experiences), we grouped participants' self-described emotional responses to irreproducibility into five categories (Fig 3). The most common response was feelings of self-doubt or loss of self-confidence (29%). These responses align with the large body of research on the imposter phenomenon in higher education [37,38]. Several participants described their experiences specifically in terms of impostorism. One participant, recalling the moment his PI decided to move him off a project after several months of being unable to produce an expected result, said: "That was probably the hardest part. That was basically when I felt like a failure and then the imposter syndrome kicked in and made me feel like I had no business being a scientist" (1054). Other participants described their reactions in terms of annoyance/frustration (26%), depression/demotivation (17%), or anxiety/panic (7%).

We categorized 21% percent of participants as reacting with equanimity to their experience with irreproducibility. These participants were unsurprised to encounter irreproducibility and did not view failures to replicate or null results as problematic. In some cases, this was because participants felt protected from pressures to publish because they had already had success in other projects; in other cases, it was because they felt that all results should be valued equally if they were obtained using rigorous methods. As one participant put it, "The goal of science is to make sure you're analyzing your data correctly, you are using the most rigorous methods, you

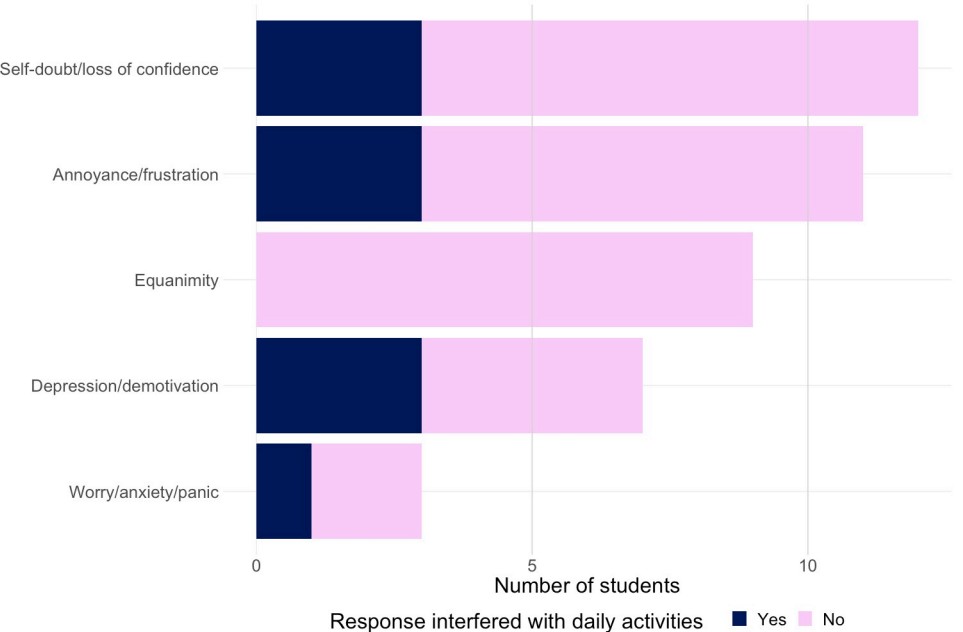

**Fig 3. Self-described emotional responses to irreproducibility experiences (n = 37 participants, n = 42 experiences).** Experiences where the participant's emotional response was strong enough to interfere with sleeping, eating, work performance, or relationships are shaded in blue.

have a good sample size, and whatever. If you follow all those steps and you're careful in methodology, whatever you find isn't failure by any means, right? It is just another result" (1008).

In addition to disease-specific symptoms, the DSM-5 criteria for generalized anxiety disorder and major depressive disorder both require that a patient's symptoms cause "clinically significant distress or impairment in social, occupational, or other important areas of functioning" to make a diagnosis [39]. We examined participants' transcripts for indications that their emotional responses to their irreproducibility experience were strong enough to interfere with activities such as sleeping, eating, performing job tasks, or relationships with family and friends, and found evidence of this in 24% of experiences. Participants whose responses to irreproducibility interfered with their daily activities were found across all categories of emotional response except equanimity. Box 1 provides quotes illustrating the range and severity of impacts on daily living that participants described in their interviews.

### Box 1. Representative quotes illustrating impact of irreproducibility experiences on participants' daily activities

Just having this issue present last month, it really messed up a lot of things personally for me, my sleep schedule, my attitude to work. I'd do all my experiments, then I'd come home and I was like, "I really don't feel like doing the data analysis today," or "I really don't feel like doing the paper writing" (2003).

I had constant headaches. My stress levels were very high. It made it hard to focus on school because I was constantly in the back of my mind trying to figure out what was wrong with my experiment, how to make it work. . . My sleeping became impaired because I was stressed thinking about it at night and stuff (1038).

[I had] anxiety and depression. It was not great. . . There were a couple of points in the middle there where it became functionally obstructive. Work productivity would go down because of that, and then you just start assuming that the projects aren't going to work. You start assuming that the experiments don't work and aren't going to work, and then it becomes a vicious cycle (1005).

I did have some bad anxiety at the time and borderline depression. That really fed into it and made it that much worse, just the failure to replicate and not being able to get things done in the lab. . . I was very lost and didn't have a lot of motivation. It was really hard to want to keep working on that project or figuring out what I'm going to do. I don't think I got more than a couple of things done in the day. When I'm more productive I can get ten things done but then it was like maybe two (1054).

I am a person that keeps a pretty level head and I never really let my emotions get the best of me. I never thought I would have said it, but looking back on it, I was actually depressed. I wasn't eating correctly. I wasn't sleeping correctly. Like I said, I'm a pretty strong person mentally, but I remember one time after a meeting with my PI and being told what I already knew, but then also having my apparent inadequacy in my face, I had to go to a single occupancy bathroom. I got in the corner and just broke down crying. I was having constant conversations with my girlfriend at the time. . . and she could attest to this that I was actually very close to quitting grad school. It was very impactful (1021).

I was super depressed after [the failure to replicate] for a solid three or four months. . . Clearly, I've got a predilection to that sort of thing, but still, those were definitely triggering events (1007).

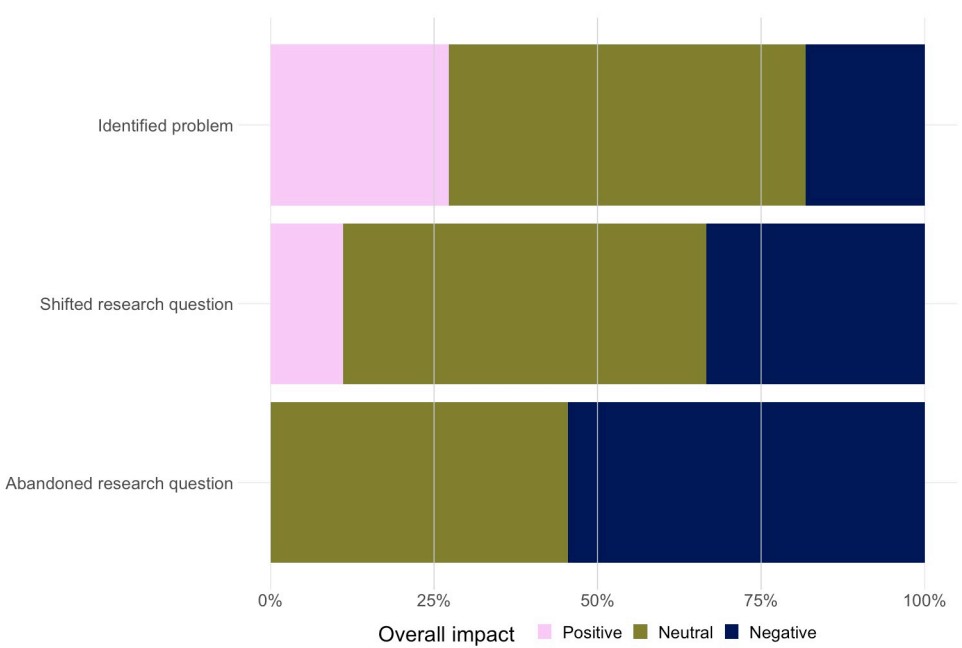

**Fig 4. Overall impact of irreproducibility experiences broken down by eventual outcome.** Experiences are separated into instances where the participant: 1) identified the source of their anomalous results, 2) shifted their research question so that the anomalous results were no longer relevant, or 3) abandoned the line of research without identifying the source of the problem. The proportion of experiences in each category resulting in overall positive/ neutral/negative impacts on the participant is displayed on the x-axis.

Finally, we categorized participants' descriptions of the conclusion to their irreproducibility experience and the overall impact of the experience on their mental health, career progression, and enthusiasm for science (Fig 4). Participants described 53% of experiences as having no strong positive or negative impacts, 31% as having overall negative impacts, and 17% as having overall positive impacts. Participants reporting positive impacts generally described their irre-producibility experiences as difficult at the time but beneficial in the long term because they increased their confidence in their ability to overcome obstacles. As one participant put it, "I'd say it improved my mood, enthusiasm. It just made me feel like a competent graduate student, being able to not just run experiments but do some problem-solving" (1040).

Fifty-three percent of students framed the conclusion of their stories in terms of having identified their problem, as in the case of the student quoted above. This category included participants who were eventually able to replicate the finding by adjusting the protocol, reagents, or experimental conditions. It also included participants who were not able to repli-cate the original finding but were able to come to a reasonably definitive conclusion about why this was the case (e.g., sex or species differences, the original result was a false positive). Twenty-one percent of students reported that they had shifted their research question because of their failure to replicate, which included situations where students shifted their research question to study the source of the irreproducibility itself. In other instances, students shifted to studying a different drug or phenotype that allowed them to pursue a similar line of research but avoid the problem of the anomalous results. Finally, 26% of students reported abandoning the line of research related to their irreproducibility experience without identifying the source of the anomalous results. Pairwise comparisons using Dunn's test indicated that participants who had abandoned a line of research had significantly lower scores for overall impact

compared to those who believed they had identified their problem (p = .036), although it should be noted that this is an exploratory analysis that was not hypothesized in advance.

## Discussion

Our findings on irreproducibility experiences align with data from prior surveys, which found that 49% of graduate students in cancer biology and 72% of members of the American Society for Cell Biology had failed to replicate a finding from the published literature [31,32]. While our sample size is small compared to these prior studies (n = 80 versus n = 1159 and n = 869), our response rate is high (63% versus 11% and 15%), and our survey assessed internal replications and replications of unpublished studies in addition to replications of published studies. The high response rate that we saw may be an effect of the Covid-19 pandemic since the survey was conducted during the initial months of lockdown when students did not have access to their lab spaces (and therefore had fewer work tasks to complete). The pandemic may also account for the low percentage of women participating in the interview phase of the study compared to the survey, since time use studies have shown that women spent more time on caregiving tasks during the pandemic than men [40].

Importantly, we paired our survey data with qualitative interviews that provided in-depth information about how participants interpreted irreproducibility experiences and how they believed those experiences impacted their mental health. From this qualitative data, we are able to make connections between the experience of failing to replicate a result and mental health. One-quarter of our participants described their irreproducibility experience as directly impacting their ability to sleep, eat, perform tasks at work, or maintain relationships outside of work, suggesting that these experiences may trigger clinically significant mental health symptoms in some graduate students. Future research could use the coding scheme derived from this exploratory qualitative study to design survey questions to explore quantitative relationships between irreproducibility and mental health in larger samples, or to investigate whether responses to irreproducibility are related to personality traits or levels of self-efficacy. A handful of recent studies have shown relationships between graduate student mental health and scientific/research self-efficacy [41–43], and considering irreproducibility experiences as events that contribute to (or detract from) the development of scientific/research self-efficacy may aid in better theorizing this relationship.

A limitation of our work is that we only studied graduate students supported by National Institutes of Health (NIH) training grants. This excluded international students and students with less robust funding packages, both populations which may have a higher risk of poor mental health because of stress related to their employment situation. Prior research has found that international students report feeling acute pressure to produce results that support their advisors' favored hypotheses because their visa status is tied to their student position [31]. Additionally, those training grant program directors who agreed to participate in our study may have had a stronger existing interest in graduate student mental health and may have therefore created a more supportive training environment for their students. We also recruited more men than women for both the survey and interview components of the study, which may have led to underestimates of mental health problems since women have twice the lifetime rates of depression and most anxiety disorders [44]. Taken together, these limitations suggest that the risk of mental illness in our study population was likely lower than in the broader population of doctoral STEMM students studying in the United States, and our comparison with the HMS survey sample from the same year shows significantly higher rates of probable depression in the nationally representative sample. Our findings may therefore underestimate the impact of failures to replicate on the mental health of biomedical graduate students because

our study population had a greater number of protective factors such as stable income, supportive training environments, and male sex.

An additional limitation is that we could only measure current mental health status using validated screening tools and relied on participants' retrospective self-reports of how their prior irreproducibility experiences impacted their mental health at the time. These experiences occurred ranging from as recently as a few weeks prior to the interview to as long as several years ago, which may result in different degrees of accuracy in participants' recollections. Future studies could attempt to study the intersection of irreproducibility and mental health in real time, taking measurements of current mental health while students are experiencing a failure to replicate and collecting longitudinal interview data to see how participants' interpretations change over time.

Our study adds to the existing literature on student mental health by suggesting a mechanism through which experiencing irreproducibility could lead to poor mental health. We hypothesize that graduate students' tendency to default toward interpreting anomalous findings as indications of a lack of personal skill could be considered a type of cognitive distortion that may contribute to the onset or perpetuation of anxiety, depression, and impostor phenomenon. Participants in our study recalled that they tended to consider other possible explanations for their anomalous results only after intervention from a mentor or many repetitions of the same experiments. Participants' initial assumptions about the source of the problem were not evenly distributed across categories; in a strong majority of cases (82%), participants initially believed they were at fault. Our data further suggest that this was not because student error was the most probable explanation—at the time of the interviews, only one participant believed their anomalous results were due to their own error, and 38% had shifted to believing that the original results were incorrect. While it is not unreasonable for scientists who are new to a technique first to ask whether they have performed the technique correctly, the discrepancy we identified between participants' initial and current explanations of their anomalous results suggests that students are initially biased toward the "student error" explanation. This potential bias has implications for both graduate students' mental health and the progress of science. Misattributions about the source of irreproducibility represent a delayed or missed opportunity to refine the scientific record. If graduate students default to blaming themselves and question the strength of the original finding only after months or years of further experimentation, this may decelerate the rate at which published findings are evaluated and corrected.

Our research adds to the literature on STEMM graduate education by providing a novel explanation for why the odds of attrition and mental health problems are higher in the beginning phases of the PhD [17,45,46]. While failing to reproduce a result might be a nearly universal experience for biomedical scientists, our data suggest that its impact depends on when it occurs in the student life course. Participants in our study noted that irreproducibility experiences early in their career were more impactful because they had not yet had opportunities to grow confident in their research skills or to demonstrate their capacities to their advisors through the successful completion of projects. Considering irreproducibility experiences alongside already identified risk factors such as departmental culture, the student-advisor relationship, and the perceived job market [45,47–49] may enhance existing explanations for why attrition rates and poor mental health are higher in the first years. In addition, our preliminary data showed that one-third of participants recalled irreproducibility experiences from their undergraduate education, suggesting that irreproducibility early in one's career can have a lasting impact. Understanding how undergraduate students process and respond to irreproducibility may aid in understanding who decides to pursue a career in science.

Our research also implicates irreproducibility experiences in the development of student-advisor relationships. Mentorship is a critical factor for STEMM student retention and success [50] and a protective factor for student mental health [42,51]. Our study is in line with this large body of research, showing that good mentorship can shift students away from feelings of impostorism and towards alternative interpretations of irreproducibility. Problems may arise, however, when advisors are also biased towards the "student error" explanation for anomalous results, which may reinforce students' feelings of impostorism and/or create conflict in the advisor-student relationship. In our study, some participants recalled that their advisors continued to assert that they were at fault even after the participants themselves had arrived at an alternative explanation. As one participant put it, the feeling of "not being believed" (1047) generated distress and damaged their relationship with their advisors. Participants reported feeling angry and helpless when advisors refused to order new reagents because they didn't believe the student's assertions or pushed the student to keep repeating experiments that the student believed were destined for failure. Participants also reported that being forced to abandon a project by their advisor when they felt their problems were solvable generated feelings of self-doubt and lack of agency. It appears, therefore, that discrepancy in the interpretation of failures to replicate in either direction can strain the student-advisor relationship, putting the student at increased risk of poor mental health.

Psychoeducational interventions may effectively counter the impact of irreproducibility experiences on graduate student mental health. The literature on the efficacy of psychoeducational interventions for mental health promotion is mixed [52,53]. However, several meta-analyses have found psychoeducation about stressors in the college student life course to be effective in reducing anxiety in student populations [54–57], particularly when these interventions are embedded in a "supervised practice" model where students practice the skills introduced under the supervision of a mentor [56]. A psychoeducational module on irreproducibility could inform new graduate students that failing to replicate a result is a common experience, that interpreting this experience as a referendum on their experimental skill can trigger feelings of impostorism, and that they should aim to consider all possible explanations—including that the original result might be incorrect—as they conduct follow-up experiments. Supervised practice of these skills could take place through the existing apprenticeship model of graduate education, where PIs could reinforce messages about the universality of irreproducibility experiences and encourage students to consider alternative explanations when meeting with students to discuss their experimental results.

Finally, we hope these findings will help program directors, advisors, and graduate student peers better assess which students might be at higher risk of poor mental health and refer those students to appropriate services. Recent research has shown that over half of life science graduate students with depression revealed their mental health status to their advisor and almost three quarters revealed it to a peer [58]. Our study suggests, however, that graduate students may not always label mental health problems as such. We found that relatively few participants (7%) described their experiences in terms of anxiety, which is noteworthy because of the high prevalence of probable anxiety disorders seen in our survey, the HMS survey data, and other studies of graduate students [19,59]. The gap between the prevalence of anxiety disorders and the low rates of self-described anxiety in our study suggests that some students with anxiety disorders may articulate their experiences in other terms, such as self-doubt or frustration. Advisors and program managers should be aware of the varied vocabulary that students might use to express distress when considering who to refer for evaluation of anxiety disorders. More generally, identifying early experiences of irreproducibility as a risky stage in the scientific life cycle will allow advisors and program directors to supplement existing mental health promotion programs with targeted information delivery to students when they need it most.

## Conclusion

To our knowledge, our study is the first to investigate the relationship between irreproducibility and graduate student mental health. Our data suggest that failing to replicate a prior finding is a common, almost universal experience in biomedical research—only one participant in our study reported no experience with any of the scenarios we described. Students who experienced irreproducibility overwhelmingly defaulted to assuming that they made an error or lacked the necessary skill to reproduce the result, although intervention from a mentor or further experimentation did shift these interpretations over time. For some students, these experiences triggered mental health symptoms that were pronounced enough to interfere with daily life. Students who felt protected from publication pressures and viewed the published literature with skepticism were more able to react with equanimity to a failure to replicate. These differential responses have important implications both for graduate student well-being and for the scientific record, since a tendency to dismiss irreproducibility as student error may delay the pace at which corrections are made.

## Supporting information

**S1 File. Survey.**
(PDF)

**S2 File. Interview guide.**
(PDF)

**S3 File. Code book for analysis of interview data.**
(PDF)

**S4 File. COREQ checklist.**
(PDF)

## Acknowledgments

We thank Raphael Ellis for his assistance with data collection and literature review, Zach Bernhart for his assistance with data collection and analysis, and Alexis Smiezek for her assistance with data management.

## Author Contributions

**Conceptualization:** Nicole C. Nelson.

**Data curation:** Nasser Lubega, Abigail Anderson.

**Formal analysis:** Abigail Anderson, Nicole C. Nelson.

**Funding acquisition:** Nicole C. Nelson.

**Investigation:** Nasser Lubega, Abigail Anderson.

**Methodology:** Nicole C. Nelson.

**Project administration:** Nicole C. Nelson.

**Supervision:** Nicole C. Nelson.

**Validation:** Abigail Anderson.

**Visualization:** Nicole C. Nelson.

**Writing – original draft:** Nasser Lubega, Abigail Anderson, Nicole C. Nelson.

**Writing – review & editing:** Nasser Lubega, Abigail Anderson, Nicole C. Nelson.

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
