## [Editor Report · Decision Letter 0]

29 Nov 2022

PONE-D-22-24719

Experience of irreproducibility as a risk factor for poor mental health in biomedical science doctoral students: A survey and interview-based study

PLOS ONE

Dear Dr. Nelson,

Thank you for submitting your manuscript to PLOS ONE. After careful consideration, we have decided that your manuscript does not meet our criteria for publication and must therefore be rejected.

I am sorry that we cannot be more positive on this occasion, but hope that you appreciate the reasons for this decision.

Kind regards,

Nabeel Al-Yateem, PhD

Academic Editor

PLOS ONE

Additional Editor Comments:

I could not find reviewers to review the manuscript.the topic and the study is not so convencing
---

## [Decision Letter · Decision Letter 1]

29 Mar 2023

PONE-D-22-24719R1

Experience of irreproducibility as a risk factor for poor mental health in biomedical science doctoral students: A survey and interview-based study

PLOS ONE

Dear Dr. Nelson,

Thank you for submitting your manuscript to PLOS ONE. After careful consideration, we feel that it has merit but does not fully meet PLOS ONE’s publication criteria as it currently stands. Therefore, we invite you to submit a revised version of the manuscript that addresses the points raised during the review process.

ACADEMIC EDITOR: Please insert comments here and delete this placeholder text when finished. Be sure to:

Indicate which changes you require for acceptance versus which changes you recommendAddress any conflicts between the reviews so that it's clear which advice the authors should followProvide specific feedback from your evaluation of the manuscript

We look forward to receiving your revised manuscript.

Kind regards,

Diego A. Forero, MD; PhD

Academic Editor

PLOS ONE

Journal Requirements:

2. We note that you have referenced (ie. Bewick et al. [5]) which has currently not yet been accepted for publication. Please remove this from your References and amend this to state in the body of your manuscript: (ie “Bewick et al. [Unpublished]”) as detailed online in our guide for authors

Additional Editor Comments (if provided):

Reviewers' comments:

Reviewer's Responses to Questions

**Comments to the Author**

1. If the authors have adequately addressed your comments raised in a previous round of review and you feel that this manuscript is now acceptable for publication, you may indicate that here to bypass the “Comments to the Author” section, enter your conflict of interest statement in the “Confidential to Editor” section, and submit your "Accept" recommendation.

Reviewer #1: (No Response)

Reviewer #2: (No Response)

2. Is the manuscript technically sound, and do the data support the conclusions?

Reviewer #1: Partly

Reviewer #2: Yes

3. Has the statistical analysis been performed appropriately and rigorously? 

Reviewer #1: Yes

Reviewer #2: Yes

4. Have the authors made all data underlying the findings in their manuscript fully available?

Reviewer #1: Yes

Reviewer #2: Yes

5. Is the manuscript presented in an intelligible fashion and written in standard English?

Reviewer #1: Yes

Reviewer #2: Yes

6. Review Comments to the Author

Reviewer #1: Overall, it is a great research idea which tried to connect irreproducibility experiences with mental health, I personally think this is a great topic, but here are some issues I wish the authors should consider.

1. To help readers understand the nature of survey questions, it will be more helpful if the authors could give some examples of the survey questions.

2. I noticed that the literature was not well-updated, the latest were 2021, two years passed and there are many relevant literatures published, so I suggested the authors update your literature review to make the research sound and solid.

3. It seems that the authors intended to use mixed methods to illustrate the connection between irreproducibility experiences and mental health, I think the qualitative part is well constructed but there are still lots things to do for quantitive analysis, at least some correlation analysis or simple linear regression to show if there are quantitative links between them.

4. In “Materials and Methods”, it will be much better if the authors can list in what ways you analyzed your data, especially the survey results, if you just list the percentage of all the survey results, it seems difficult to prove there is a connect between irreproducibility experiences with mental health.

5. To make a good use of quantitative data, the survey results, I suggested to run correlation tests and multivariate regressions on the mental health survey congragates and Responses to survey questions about potential irreproducibility scenarios, which somewhat tell us if there is an association between them, and percentage won’t do the work.

Reviewer #2: The manuscript presents an informative study that seeks to discover the relationship between the experience of not being able to replicate research results (in the laboratory), and mental health/well-being in a sample of biomedical science graduate students. The study was conducted in a group of NIH graduate fellowship recipients. The study uses a combination of quantitative information from wellness surveys, mental health (depression and anxiety), and qualitative data from semi-structured interviews with a subsample of students who reported more than one irreproducibility-related experience. As external validation of the survey results, the researchers compared their results with those collected with the HMS survey in the same year of the study.

The results show that most students have experienced problems of replication of results (their own, related to their work environment or external to their work environment). These reproducibility problems affected the well-being of most students, and the impact was related to their perceived self-confidence in the quality of their own work, the experience in the field of study and the support received by their mentors.

Comments:

The study was conducted in 2020, during the most critical period of the covid 19 pandemic. The authors believe that this phenomenon may have had some confounding effect on some of the measurements or interview results.

Although the response rate was moderate/high, since the sample is not random, there is a possibility that in students who did not participate in the study (but were invited) there was a higher proportion of people with emotional distress or previous negative experiences. Is there any way to verify this hypothesis in the study population?

Is there any possible explanation as to why the proportion of women who agreed to participate in the interviews was lower? What strategies do you recommend could be followed to encourage more women to participate in the interviews?

What was the time available to answer the 3 surveys? Did the researchers conduct internal consistency tests to verify that the quality of the responses was not affected by completing several surveys simultaneously?

In the methods section, when discussing the HMS test, a sentence could be included clarifying that the test was used as external validation of the study data. At first glance the reader may understand that it is an additional survey that was applied in the study.

Although most of the results describe qualitative information. It would be useful to include, in the methods section, a short description of the statistical analyses used to evaluate the quantitative data.

The people who coded the results took part in the interviews. This information is not clear in the document.

The discussion suggests that after living experiences related to reproducibility (post hoc) the participants developed experience or defense mechanisms that allowed them to reinterpret the lived events. However, it is also possible that some of these defense mechanisms are related to their previous levels of self-efficacy, or personality traits.

About 20% of the participants showed "indifference" which could be interpreted as 1) "apathy" or alternatively as 2) "resilience" to the lived experience. Which of these responses would be more related to the participants' coping mechanisms for the situation.

For the comments in box 1, it would be useful to clarify whether some of them come from the same person or all of them come from different participants.

For the reader, it might be useful to have a concluding paragraph at the end of the discussion

Please consider adjusting the color scheme in Figure 1 to make it easier for readers to see the categories.

7. PLOS authors have the option to publish the peer review history of their article (what does this mean?). If published, this will include your full peer review and any attached files.

Reviewer #1: No

Reviewer #2: No

---

## [Author Response · Author response to Decision Letter 1]

23 Aug 2023

Editor

1. Ensure that the manuscript meets PLOS ONE requirements for file naming.

We are guessing that this request refers to the citations rather than the file names—we did not find any errors in the file names, but we did find that our citations had not been generated using the proper style. We have redone the citations/bibliography and removed the Zotero field codes to prevent potential problems after uploading. Additionally, we have checked the image files using PACE and changed the names of the supplementary files to reflect the addition of one new file.

2. Remove Bewick et al from reference list because it is unpublished.

There is no reference to a paper by Bewick et al in our manuscript.

3. Review reference list and justify citations to retracted papers.

As far as we are aware, there are no citations to retracted papers in our manuscript. If the editor is aware of issues with some of the papers cited, please let us know which papers and we will happily revise the manuscript.

Reviewer One

1. Give examples of the survey questions.

We have added a new supplementary file (S1 File) that contains the survey in its entirety.

2. Update the literature review.

We have used several recently published systematic reviews/meta-analyses/meta-syntheses to more comprehensively describe the relevant literatures, particularly the literature on graduate student mental health, which the reviewer rightly notes has been very active in the past few years (likely owing to the mental health impacts of the Covid-19 pandemic).

3. Perform additional quantitative analyses, such as correlations or linear regressions. Specifically, perform correlation/multivariate regressions on the mental health measures and questions about irreproducibility in the survey data.

We have elected not to perform these analyses because we don’t believe that looking for relationships between current mental health and past experiences of reproducibility would be appropriate given the data we have collected. We did not collect data describing when participants experienced irreproducibility, only their lifetime to date experiences of irreproducibility. If we were to perform the analyses suggested, we would be comparing irreproducibility events with very different recency to current mental health with no way of controlling for time elapsed. We collected current mental health data primarily for the purposes of comparing our population to reference populations to assess whether we had a significantly lower/higher frequency of current mental illness in our population. We have added additional detail about this rationale in the new section in the materials and methods described below.

4. Describe the analysis of the survey data in the materials and methods section.

A new section on the analysis of the survey data has been added to the materials and methods section. 

Reviewer Two

1. Describe the potential impact of the pandemic on the study.

We now describe two potential impacts of the pandemic on the study: we may have experienced a high response rate due to participants’ inability to access their lab spaces (and therefore having fewer work tasks to complete), and we may have experienced a lower response rate from women because of women’s increased participation in caretaking labor during the pandemic. We also clarify that the pandemic almost certainly had an impact on participants’ mental health, and therefore it is particularly important that we are comparing current mental health data from our population to data taken from the HMS survey during the same time period. 

2. Account for possible response bias in the study.

We now clarify in the methods section that the reason for comparing our population to the HMS study population is because the HMS survey had complete data on the populations sampled and was therefore able to control for response bias. We did not have access to data for non-responders and therefore could not make corrections for response bias, but we assume that similar response biases are operative in our study as were seen in the HMS study. We now show both the raw and weighted data from the HMS survey in Table 1 to make those biases clear, and explicitly discuss response bias in the manuscript. 

3. Explain why the percentage of women who participated in the study was lower.

See the response to question R2.1 above

4. Describe the time needed to answer the survey, and any internal consistency tests performed to verify that the quality of the responses was not impacted by answering three surveys simultaneously.

We have added information on the survey duration to the methods section. We did not perform any internal consistency tests because the total time taken to complete the survey was less than six minutes. As mentioned above, we have also now included the survey as an appendix so that readers can get a better sense of the total length of the survey.

5. Clarify that the HMS study data was used for external validation in the methods section.

We have clarified this in the methods section.

6. Describe the statistical analyses used to evaluate the quantitative data in the methods section.

We have added a new section to the materials and methods, as also suggested by R1 above.

7. Clarify that the coding was performed by the same researchers who took part in the interviews.

We have clarified this in the qualitative data analysis methods section and clarified that the interview transcripts were identified by participant ID only during the qualitative data analysis phase.

8. Variation in how participants respond to/interpret failures to replicate may be related to personality traits or prior levels of self-efficacy.

We think this is an excellent suggestion for future research and have added it to the discussion section. 

9. Consider whether indifference is better interpreted as apathy or as resilience to the experience of failing to replicate a result.

We thank the reviewer for this helpful comment—we do not think that what we initially classified as “indifference” is best interpreted as apathy, and so have changed the descriptor for this category to “equanimity.” We believe this better captures the spirit of this category and helps avoid misreading this category of emotional response as apathy or anhedonia, which would fall under the category we have termed “depressed/demotivated/tired.”

10. Clarify whether the interview excerpts in Box 1 come from different participants.

Each quote in Box 1 comes from a different participant. We have added participant ID numbers to all interview excerpts throughout the manuscript to clarify this.

11. Add a concluding paragraph to the discussion.

We have added a concluding paragraph.

12. Adjust the color scheme in Figure 1.

We have changed the color palette from viridis to scico, which is another scientifically-derived color map that has been designed for perceptual uniformity, to be interpretable by readers with different forms of color blindness, and to render properly in greyscale.

---

## [Decision Letter · Decision Letter 2]

17 Oct 2023

Experience of irreproducibility as a risk factor for poor mental health in biomedical science doctoral students: A survey and interview-based study

PONE-D-22-24719R2

Dear Dr. Nelson,

We’re pleased to inform you that your manuscript has been judged scientifically suitable for publication and will be formally accepted for publication once it meets all outstanding technical requirements.

Kind regards,

Diego A. Forero, MD; PhD

Academic Editor

PLOS ONE

Additional Editor Comments:

The authors have included the suggestions previously made by the reviewers.

Additional Journal Comments:

Please note that the reviewer has included a minor request for clarification re: the results section. Please amend this prior to submitting your final version of the manuscript.

Reviewers' comments:

Reviewer's Responses to Questions

**Comments to the Author**

1. If the authors have adequately addressed your comments raised in a previous round of review and you feel that this manuscript is now acceptable for publication, you may indicate that here to bypass the “Comments to the Author” section, enter your conflict of interest statement in the “Confidential to Editor” section, and submit your "Accept" recommendation.

Reviewer #2: All comments have been addressed

2. Is the manuscript technically sound, and do the data support the conclusions?

Reviewer #2: Yes

3. Has the statistical analysis been performed appropriately and rigorously? 

Reviewer #2: Yes

4. Have the authors made all data underlying the findings in their manuscript fully available?

Reviewer #2: Yes

5. Is the manuscript presented in an intelligible fashion and written in standard English?

Reviewer #2: Yes

6. Review Comments to the Author

Reviewer #2: Minor comment

For the reader it may not be immediately obvious that the four digit number, at the qualitative results section, correspond to a participant. This could be clarified in the methods section or within the parentheses (participant #xxxx)

7. PLOS authors have the option to publish the peer review history of their article (what does this mean?). If published, this will include your full peer review and any attached files.

Reviewer #2: No

---

## [Editor Report · Acceptance letter]

23 Oct 2023

PONE-D-22-24719R2 

Experience of irreproducibility as a risk factor for poor mental health in biomedical science doctoral students: A survey and interview-based study 

Dear Dr. Nelson:

I'm pleased to inform you that your manuscript has been deemed suitable for publication in PLOS ONE. Congratulations! Your manuscript is now with our production department. 

Kind regards, 

on behalf of

Dr. Diego A. Forero 

Academic Editor

PLOS ONE